# Functional Outcomes and Shoulder Instability in Reconstruction of Proximal Humerus Metastases

**Alessandro El Motassime** [1], **Cesare Meschini** [1,2], **Doriana Di Costa** [1,*], **Giuseppe Rovere** [1,2], **Maria Rosaria Matrangolo** [1], **Fernando De Maio** [3], **Pasquale Farsetti** [3] 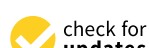, **Antonio Ziranu** [1,2], **Giulio Maccauro** [1,2] and **Raffaele Vitiello** [1,2]

1   Department of Orthopaedics, Fondazione Policlinico Universitario Agostino Gemelli IRCCS, 00168 Rome, Italy
2   Department of Orthopaedics, Fondazione Policlinico Universitario Agostino Gemelli IRCCS, University Cattolica del Sacro Cuore, 00168 Rome, Italy
3   Department of Clinical Science and Translational Medicine, Section of Orthopaedics and Traumatology, University of Rome "Tor Vergata", 00133 Rome, Italy
*   Correspondence: doriana.dicosta01@icatt.it; Fax: +39-06-305-1161

**Abstract:** (1) Background: Some of the goals of orthopedic surgical oncology are saving limbs and function. The humerus is the third most frequent site in primary tumors and one of the most involved sites for metastases. Prosthetic replacement with modular megaprosthesis is one of the treatment choices, but there are several types of complications, such as problems with function and pain. The aim of our study is to assess functional outcomes and shoulder instability in the reconstruction of proximal humerus metastases. (2) Methods: This is a retrospective observational study. Twenty-eight patients, with proximal humerus metastases, admitted to the department of Orthopaedics and Traumatology of our University Hospital between 2014 and 2022 were recruited. Each patient underwent resection and prosthetic replacement surgery with modular megaprosthesis. Clinical evaluation was assessed through MSTS score, WOSI index, and DASH score. (3) Results: Twenty patients were included in the study. Fairly good results, especially regarding pain, function, and emotional acceptance, were obtained in all three tests: DASH, MSTS, and WOSI. Patients who reported shoulder instability actually have worse outcomes than those who report having stable shoulders. In addition, patients with a resection >10 cm have worse outcomes than those who had a resection of 10 cm. No significant differences were found between the deltopectoral approach group and the lateral approach group. (4) Conclusions: Reconstructive surgery with megaprosthesis of the proximal humerus in patients with metastases can be considered a treatment option, especially in patients with pathological fractures or injuries with a high risk of fracture and good life expectancy. This study shows how this type of surgery affects instability, but in terms of functionality, pain, and patient satisfaction, it gives satisfactory results.

**Keywords:** shoulder; megaprosthesis; functional outcome; proximal humerus

## 1. Introduction

The goal of orthopedic surgical oncology has for many years become saving limbs and function as much as possible. Primary tumors of bone are rare, but as the population ages and treatment improves, the number and frequency of metastatic lesions are increasing. The humerus is the third most frequent site in primary tumors and one of the most involved sites for metastases [1,2]. There are several treatment options in these cases; for some lesions, curettage or grafting is a treatment option, or in the case of fractures, plate synthesis or intramedullary nailing is required [3]. In some cases, these treatments are not sufficient and it may be necessary to perform a bone resection with subsequent reconstruction. To choose the type of surgical treatment must evaluate several criteria such as life expectancy, histotype, staging, and the patient's condition [4]. It is possible to choose

between numerous types of reconstructive treatments after resection such as prosthetic reconstruction with anatomic or reverse shoulder systems or custom prosthesis, allograft-prosthesis composites, arthrodesis, the clavicula pro humeri procedure or the fibula pro humeri procedure [5–7]. According to the criteria of Capanna and Campanacci, resection with arthroplasty reconstruction is indicated in the case of pathological fractures or injuries with a high risk of fracture when the patient's survival expectancy is good (more than 6–12 months) [8]. Reconstruction with megaprosthesis may be associated with a reduction in function and shoulder instability, considered one of the most common complications in this type of surgery [1,9] and felt by the patient as a loss of comfort and function due to unwanted translation and loss of the joint relationship between the prosthesis and the glenoid [10]. These complications may arise if rotator cuff muscles or tendon insertions or the axillary nerve have to be sacrificed to obtain a satisfactory tumor margin [11–15] and can result in negative outcomes and the need for revision surgery. The aim of this study is to evaluate the functional outcome and shoulder instability of reconstructive surgery with megaprosthesis of the proximal humerus.

## 2. Materials and Methods

A retrospective observational study was performed in accordance with PROCESS guidelines. As this is an approval from the Review Board of Orthopedic and Traumatology Institute, there is no code. The approval date is the session of 22 June 2021. The study complies with national ethical standards and the Declaration of Helsinki. Each patient was given informed consent for surgery and for the collection of clinical data for scientific purposes at admission and before the surgery, according to institutional protocols.

Twenty-Eight patients were recruited in our study (13 males and 15 females; mean age: 61.5 years old) treated in our institution between 2014 and 2022. Table 1.

**Table 1.** Demographic table.

| Individual-Level Variables | N | Percent | Mean | SD |
|---|---|---|---|---|
| Age | | | 61.3 | 13.26 |
| 40–50 | 4 | 20% | | |
| 51–60 | 4 | 20% | | |
| 61–70 | 8 | 40% | | |
| 71–80 | 2 | 10% | | |
| 81–90 | 2 | 10% | | |
| Gender | | | | |
| Male | 12 | 60% | | |
| Female | 8 | 40% | | |
| Comorbidities | | | | |
| Diabetes | 11 | 55% | | |
| Thyroid pathologies | 2 | 10% | | |
| Primary tumor | | | | |
| Kidney | 8 | 40% | | |
| Breast | 4 | 20% | | |
| Lung | 2 | 10% | | |
| Brain | 2 | 10% | | |
| Lymphoma | 2 | 10% | | |
| Uterus | 2 | 10% | | |
| Total | 20 | | | |

The inclusion criteria were proximal humerus metastases, prosthetic replacement surgery with modular shoulder endoprosthesis, and pathological proximal humerus fractures, with a minimum follow up of 6 months (Mirel's score $\geq$ 9).

Mirel's scoring system is used to predict an impending fracture and thus prophylactic fixation in an elective setting to avoid debilitating complications. It consists of 4 items: site of lesion (upper limb, lower limb, trochanteric region), size of the lesion (<1/3 of the cortex,

$^1/_3$–$^2/_3$ of the cortex, >$^2/_3$ of cortex), nature of the lesion (blastic, mixed, or lytic), and pain (mild, moderate, and functional). Each of these was rated with a progressive score from 1 to 3. According to the score prophylactic fixation is highly recommended for a lesion with an overall score of 9 or greater. Almost all of our patients scored 9 or higher because they had proximal humerus metastases (score 1), lesions the size of >$^2/_3$ (score 3), mixed or lytic lesions (score 2 or 3), and functional pain (score 3) [16].

The exclusion criteria were primary tumor of the proximal humerus, proximal humerus fractures, age less than 18 years old, shoulder prosthesis revision, nailing failure revision, traumatic shoulder dislocation, and reverse shoulder prosthesis.

Regarding the fracture criteria, only the pathological ones have been considered in our study because, by definition, a pathological fracture occurs on a weakened bone, in this case, due to the presence of metastasis.

Each surgery was performed after general anesthesia in the beach-chair position. Preoperatively, antibiotic prophylaxis was administered using Cephazoline 2 g i.v. when not contraindicated. A urinary catheter was placed in all patients and removed 24–48 h after surgery. A resection surgery of the proximal humerus was performed, assessing preoperative imaging and in accordance with Enneking's resection criteria [17], and followed by replacement with silver-coated modular endoprosthesis. Minimum resection performed was 10 cm, considering the size of the smallest module of prosthesis used. In all cases, Trevira tube was used to anchor the sectioned tendons and soft tissues to increase prosthesis stability, as manufacturer's instructions. All shoulder replacement procedures were performed by the same orthopedic surgeon, experienced in orthopedic oncological surgery. Two different surgical approaches were performed, basically according to the localization of the metastases and to the level of resection; hence, the deltopectoral is an extendable surgical approach useful for bigger lesions. In 10 cases, a deltopectoral approach was used, while in 10 a lateral approach was used. The deltoid muscle and the rotator cuff tendons were preserved only if untouched by cancer or via preoperative imaging [Figure 1].

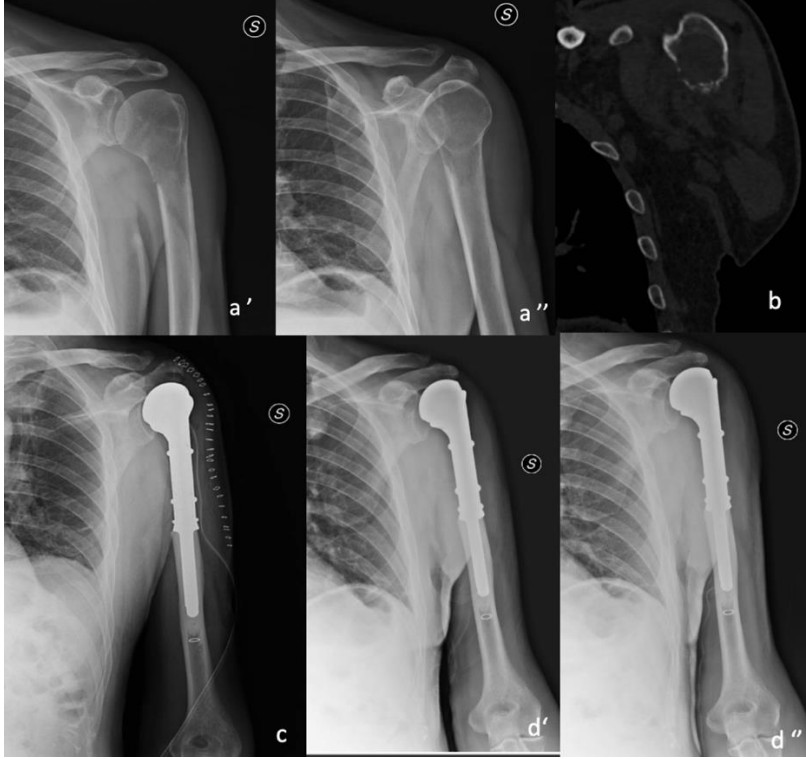

**Figure 1.** y.o. male affected by metastatic kidney tumor. Lateral approach resection was 10 cm. (**a′**,**a″**) pre-op X-ray; (**b**) pre-op ct scan; (**c**) post-op X-ray; (**d′**,**d″**) 6 months follow.

First, clinical evaluation of patients was carried out approximately 15 days after surgery. Subsequent outpatient clinical and radiographic evaluation was performed at 1, 3, 6, and 12 months. In the postoperative period, all patients were placed in an arm brace and instructed to remove it in order to perform Codman exercises and elbow flexion-extension exercises starting 15 days after surgery. Arm brace was removed at about 6 months, then patients started exercises to recover the shoulder ROM.

An assessment of the patient's pre-surgical functions was not possible due to the general clinical condition, being metastatic cancer patients or patients with pathological fractures, and because of the severe pain.

The Musculoskeletal Tumor Society rating system (MSTS) [18], Disability of Arm-Shoulder-Hand (DASH) [19] score, and Western Ontario Shoulder Instability Index (WOSI) [10] were used to evaluate function after surgery in each patient at 6 months when shoulder brace was removed.

The MSTS scoring system is used to assess function and quality of life in patients undergoing oncological surgery for musculoskeletal tumors. The MSTS score consists of six items: pain, function, emotional, external support, functional independence, and gait. Each was rated with a score from 0 to 5. A higher score indicates a better function. This score can be converted to a scale from 0 to 100 points.

The DASH score is a questionnaire that is submitted to the patient consisting of 30 questions to assess the patient's function and symptoms. Each item has five response options. The scores for all items are then used to calculate a scale score ranging from 0 (no disability) to 100 (most severe disability) [20]. Higher scores correspond to a higher felt disability by the patient.

The WOSI index is a patient-reported outcome that assesses pain, functional limitation, and quality of life in patients with shoulder instabilities. It consists of 21 items in 4 question domains, physical symptoms and pain, recreation, and work, lifestyle and social functioning, and emotional well-being. Each question has a result that is between 0 and 100. The total score ranges from 0 to 2100 (where 0 represents no deficit and 2100 represents the worst) [21].

The primary outcome was the functional outcomes after surgery (MSTS score, DASH score, and WOSI index).

Data were collected using Excel program (Microsoft, Redmond, OK, USA), and unpair (independent) *t*-test was used to confront results between groups. Statistical significance was set for $p < 0.05$.

## 3. Results

Twenty-eight patients underwent proximal humerus replacement with a megaprosthesis during our study period. Three were excluded because the megaprosthesis were implanted on primary tumor lesions or trauma patients. Five were excluded due to death before minimum follow up (6 months). The sample was thus reduced to twenty patients who met the inclusion criteria.

There were 12 females and 8 males, and the mean age was 61.3 years old ($\pm$13.26). Average follow up was 21 months.

Kidney cancer was the most common primary tumor (40%), followed by breast cancer (20%), lung cancer (10%), brain cancer (10%), lymphoma (10%), and uterus cancer (10%).

Average MSTS is 57.6% ($\pm$26.24) and shows fairly good results, especially regarding pain, function, and emotional acceptance.

Good results are confirmed by DASH score whose mean is 47.5 ($\pm$27.55).

Another score used to evaluate the results is the WOSI which resulted in an average of 950 (58.62%) ($\pm$532.29).

### 3.1. Shoulder's Stability

Notably, 12 patients (60%) answered 'extreme instability', thus corresponding to the highest score, on the question regarding the feeling of shoulder instability. In this patient,

the mean DASH was 61.4 (±22.65), the mean MSTS was 45.6 (±26.47), and the mean WOSI was 1233.3 (41.26%) (±492.95). Whereas in patients reporting feelings of stability of the shoulder, the mean of DASH was 26.7 (±16.94), the mean MSTS was 75.8 (±12.87), and the mean WOSI was 525 (84.65%) (±352.37). Significant differences emerged between these 2 groups regarding the DASH score ($p = 0.0017$), the WOSI score ($p = 0.0026$), and the MSTS score ($p = 0.0080$). Figure 2.

**Figure 2.** Stable vs. unstable patients. Reporting results of DASH, MSTS, and WOSI scores in patients who felt stable compared with those who felt instability in their shoulders.

To the best of the authors' knowledge, there is no classification for shoulder instability in patients that underwent shoulder replacement, except for reverse arthroplasty. To have a better comprehension of our patients' shoulder instability, we could try to adapt the glenohumeral joint instability classification described by Gerber et al. According to this classification, the instability of our patients' shoulders could be defined as dynamic, multidirectional instability without hyperlaxity [22].

### 3.2. Resection Surgery

Resection was >10 cm in 12 patients and 10 cm in 8 patients. The mean DASH was 59.1 (±26.61) in patients with a resection of >10 cm vs. 30.05 (±20.53) in patients with a 10 cm resection. The mean MSTS was 45.57 (±26.61) for the first group vs. 75.82 (±12.9) for the second group. The mean WOSI was 1158.33 (±484.16) for the first group vs. 637.5 (±492.23) for the second group, and the differences between these 2 groups regarding the DASH score ($p = 0.0179$), MSTS score ($p = 0.0082$), and WOSI score ($p = 0.0309$) are statistically significant.

### 3.3. Surgery Approach

Surgery was performed using the deltopectoral approach in 10 patients and using the lateral approach in 10 patients, showing no substantial differences in terms of satisfaction and shoulder stability. The mean DASH was 58.58 (±32.32) for the deltopectoral approach vs. 45.48 (±25.75) for the lateral approach ($p = 0.3294$), the mean MSTS was 54 (±31.65) for the deltopectoral approach vs. 61.34 (±22.67) for the lateral approach ($p = 0.5585$), and the mean WOSI was 920 (±610.94) for the deltopectoral approach vs. 980 (±511.86) for the lateral approach ($p = 0.8145$).

## 4. Discussion

Metastatic bone disease has a very negative influence on a patient's quality of life. A multidisciplinary approach should take place to ensure the best possible care. Surgery should only be performed in selected cases such as metastatic patients with impending or actual fractures. However, based on other parameters, such as life expectancy and amount of lesions, a combination of therapies should be evaluated [23].

To date, the standard is considered to be the wide resection which leads to the need for a complex reconstruction in order to restore the functionality of the limb for which reverse total shoulder arthroplasty is mainly used as performed on this group of patients [24].

In their study, Ebeid et al. reported patients with primary or metastatic tumors of the proximal humerus who underwent reconstructive surgery with modular proximal humerus endoprosthesis, with a mean MSTS score of 24.8 ± 1.1. In this study, this group was compared with a nail-cemented spacer reconstruction which had a mean MSTS score of 23.9 ± 1.4 [25]. In our study, patients had a mean MSTS score of 57.6% and therefore had a better functional outcome; this could be due to the use (in all surgeries) of the Trevira tube compared to the Ethibond and FiberWire sutures that Ebeid et al. used in some cases [25]. To support this hypothesis, we can take into account the D'Adamio et al. in vitro study regarding the soft tissue adhesion patterns over Trevira on modular endoprosthesis [26]. They have demonstrated in vitro how the presence of Trevira fibers around the oncological megaprosthesis gives a better anatomical reinsertion of soft tissues due to the extension of new cells (94% of them were vital cells) and their adhesion pattern [26]. Another hypothesis could be the average age of our patients (61.3 ± 13.26 years), compared with the Ebeid et al. study (33.4 ± 17.5 years), who, therefore, being older and requiring less functionality than a young adult, report a better functional outcome [25].

Teunis et al. in their systematic review on functional outcome, construct survival and complication rate in proximal humerus resection due to aggressive benign tumors or malignant tumors of the shoulder, state that in 10 studies, with a sample of 141 patients, the functional score used was MSTS with results varying between 61 and 77% [27].

In their study, Trikoupis et al. compared 2 groups of patients who underwent prosthetic shoulder reconstruction surgery after removing their bone tumors via endoprosthesis or arthroprosthesis and obtained an MSTS of 68 (±10.3) in endo patients and a DASH of 30 (±4.8) [28].

In our study, patients who underwent a resection of more than 10 cm reported worse functional outcomes than those who had a resection of 10 cm. In the endoprosthesis group, Ebeid et al. obtained a mean MSTS score of 24.8 ± 1.1, while it was 23.9 ± 1.4 in the spacer group, $p = 0.018$. These results probably depend on a selection bias as the endoprosthesis was carried out in smaller tumors with lesser muscle resection and in patients with preserved axillary nerves [25]. In accordance with our results, a larger resection could therefore influence functional outcomes.

Van der Linde et al. in a prospective cohort study have proved that the estimated minimal important change (MIC) is 14 points for the WOSI (on a scale from 0 to 100) and the smallest detectable change (SDC) is 23 points (on a scale from 0 to 100). A change in the score can be considered real and significant only if it exceeds both MIC and SDC [29], so patients with a final score lower than 77 have actually experienced a real change in shoulder instability and function. In our study, all patients who reported shoulder instability had a WOSI score below 77 (mean WOSI was 41.26).

Choosing to implant megaprosthesis was driven by prognosis, localization, and diagnosis, but great relevance has to be addressed to the possibility to use a Trevira tube which has been used primarily to anchor the tendon heads with the intention of restoring stability and reducing the rate of dislocations [26,30,31].

The meaning, utility, and importance of any membrane due to a Trevira tube or silver are still relevant today [32].

Shoulder stability is given by the deltoid, rotator cuff, and joint capsule, which are widely manipulated during the excision phase of the surgical technique, but during the

reconstructive part of the surgical technique, due to megaprosthesis design and a Trevira tube, it was possible to re-anchor every tendon that did not need removal due to tumoral involvement.

Another relevant factor to prevent complications such as infection was to use silver-coated megaprosthesis which has been implanted primarily with the intention of reducing infections, especially early ones [33,34].

Donati et al. have proved that patients treated with silver-coated implants have submitted early infection in 2.2% of cases against the 10.7% of the patients treated with standard tumor prosthesis [33]. Recent in vitro studies, confirmed by several clinical studies, have demonstrated the effectiveness of silver coatings in inhibiting or even preventing biofilm from creating on metal surfaces. On the other hand, D'Adamio et al. conducted an in vitro study of silver coating which confirmed its effectiveness in preventing surface colonization and showed antifungal properties [31].

Only 2 of our patients reported an episode of dislocation, specifically a traumatic one, which was among the patients who reported better stability (DASH 100, MSTS 0, and WOSI 1900). It is therefore possible that patients who had a greater feeling of instability, due to apprehension and therefore also limited use of the joint, may have acted more cautiously and thus obtained worse functional results but avoided this type of complication.

Our study has some limitations, such as its small sample size, due to the rare pathology and stringent inclusion criteria. Our study is retrospective, the pre-surgical function was not evaluated, and any comorbidities that could alter the results were not analyzed.

However, our study also has strengths: it was designed to minimize bias, choosing only patients with metastases to the proximal humerus; procedures were performed by a single experienced surgeon; a single design was used for the megaprosthesis; and, finally, to the best of the authors' knowledge, this is the first work that considers WOSI in patients with proximal humeral metastases.

## 5. Conclusions

Reconstructive surgery with megaprosthesis of the proximal humerus in patients with metastases can be considered a treatment option, especially in patients with pathological fractures or injuries with a high risk of fracture and good life expectancy. This study, despite its limitations, shows in fact how this type of surgery affects instability, due to the involvement of the structures that normally stabilize the shoulder, even if these are reconstructed, to the extent possible, during surgery, but in terms of functionality, pain and patient satisfaction it gives satisfactory results.

**Author Contributions:** Conceptualization and methodology, R.V.; validation, G.M., A.Z. and P.F.; formal analysis, R.V.; investigation, A.E.M., D.D.C. and G.R.; Writing—Original Draft Preparation, A.E.M., D.D.C. and C.M.; writing—review and editing, R.V.; visualization, M.R.M.; supervision, G.M., A.Z. and F.D.M. All authors have read and agreed to the published version of the manuscript.

**Funding:** This research received no external funding.

**Institutional Review Board Statement:** The study was conducted according to the guidelines of the Declaration of Helsinki, and approved by the Institutional Review Board of Orthopedic and Traumatology Institute of Università Cattolica del Sacro Cuore—Roma. As this is an approval from the Review Board of Orthopedic and Traumatology Institute, there is no code. The approval date is the session of 22 June 2021.

**Informed Consent Statement:** Informed consent has been obtained from the patient(s) to publish this paper.

**Data Availability Statement:** The datasets used and/or analyzed during the current study are available from the corresponding author upon reasonable request.

**Conflicts of Interest:** The authors declare no conflict of interest.

**Abbreviations**

MSTS: Musculoskeletal Tumor Society rating system; DASH: Disability of Arm-Shoulder-Hand score; WOSI: Western Ontario Shoulder Instability Index.

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
