# Peer review of "Functional Outcomes and Shoulder Instability in Reconstruction of Proximal Humerus Metastases"

_curroncol, doi:10.3390/curroncol30040272_

Round 1

Reviewer 1 Report

Interesting article on a hitherto unknown topic. My comments: Itroduction - well written. Materials and methods: Please provide details of the statistical analysis program. Please describe in detail the classification of degrees of shoulder instability. Results and discussion: Please describe why different surgery approach were chosen?

Author Response

Dear Sir/Madama,

Thanks for reading and finding our article interesting, we have revised the article as you suggested.

Regarding the classification of degrees of shoulder instability, there’s no a specific classification in patient that underwent shoulder replacement except for inverse arthroplasty, which is not the subject of our study.

The different surgical approach depends on the size of the metastases, in fact the deltopectoral approach being, by definition, an extensible approach is used in larger lesions.

There is no specific classification for proximal humerus metastases. In our study we use Mirel’s scoring system to predict impending fractures and thus the possibility of prophylactic surgical treatment.

Among the inclusion and exclusion criteria, however, we consider fractures, in particular only those that are defined as pathological; the mere presence of a fracture of the proximal humerus due to trauma or other causes that are not due to the presence of a metastasis were not taken into account.

As suggested, we included a demographic table.

Our study being retrospective does not have a comparison with the same instruments we used in the study such as DASH, WOSI and MSTS. Furthermore, being our patients in a metastatic stage of oncological disease and usually in severe pain, they did not have optimal conditions to accurately assess pre-surgical function.

In our study, as written, our patients had a mean follow-up of 21 months. We took 6 months as cut-off because it’s at the end of this period that the patients remove the arm brace and start the physiotherapy.

As suggested, we have replaced some expressions with some more appropriate for a scientific article.

Reviewer 2 Report

Dear Authors,

This is a good study of great importance but there are a few handicaps that I would like to bring out here.

1. In the materials and methods section the authors have merely mentioned the proximal humeral metastasis as one of the inclusion criteria, which is quite insufficient.

Please explain

a-the whole extent of proximal humeral metastasis

b-type of metastatic lesions seen

Please use any valid classification system to address this proximal humerus metastasis

2. It has also been noted that proximal humerus fracture is taken as one of the inclusion criteria for the study. To get the readers an adequate knowledge of the fractures please include any valid classification to describe the above fractures. I believe this is quite crucial in planning the surgery.

3. Please include demographic tables of the patients in the materials and methods section.

4. Please Include comorbidities such as diabetes and thyroid status along with the above table 

5. I'm afraid the pre-surgical function of the patients is not listed in the article without which post-surgical functional assessment may not be accurate. Hence please include the pre-surgical function of the patients also.

6. I believe the follow period of six months is too short of arriving at the final conclusion as we have to give time to detect the recurrence of metastasis/ failure of the prosthesis/ deterioration of functional status. Hence the follow-up period of six months is totally not acceptable.

7. Line 211 uses the word "demolition''.Kindly replace it with an appropriate scientific word/phrase.

8.LIne 212-uses the phrase "Thanks to". Kindly avoid such casual phrases in a scientific article.

9. Line 231 uses the phrase" as far as". Please replace it with an appropriate phrase.

I look forward to your early response.

Thank you.

Author Response

(The authors gave the same response as above.)

Round 2

Reviewer 2 Report

Thank you for your inputs.